# Clinical Impact of Olaparib for Platinum-Sensitive Recurrent Ovarian Cancer

**DOI:** 10.3390/diseases13020051

**Published:** 2025-02-10

**Authors:** Hiroaki Ishida, Megumi Manrai, Akiko Takashima, Hiroki Egashira, Mizuki Nonaka, Hideaki Shimada

**Affiliations:** 1Department of Obstetrics and Gynecology, Toho University Medical Center Sakura, 564-1 Shimoshizu, Sakura City 285-8741, Chiba, Japan; man-megu@sakura.med.toho-u.ac.jp (M.M.); takashima-04@sakura.med.toho-u.ac.jp (A.T.); hiroki.egashira@med.toho-u.ac.jp (H.E.); mizuki.motohashi@med.toho-u.ac.jp (M.N.); 2Department of Gastroenterological Surgery and Clinical Oncology, Graduate School of Medicine, Toho University, 6-11-1 Omori Nishi, Ota-ku 143-8541, Tokyo, Japan; hideaki.shimada@med.toho-u.ac.jp

**Keywords:** ovarian cancer, olaparib, platinum-sensitive recurrent ovarian cancer, CA125

## Abstract

**Background/Objectives**: Olaparib, a poly ADP ribose polymerase inhibitor, has been effective in prolonging progression-free survival in platinum-sensitive recurrent ovarian cancer. The clinicopathological factors that predict a favorable prognosis remain unclear. Therefore, we retrospectively analyzed the prognostic effect of clinicopathological factors in the patients treated with olaparib for platinum-sensitive recurrent ovarian cancer. **Methods**: A total of 16 patients were treated with olaparib from 2018 to 2023. We categorized these patients into the responder (five cases who had not relapsed within 2 years) and non-responder groups (11 cases who had relapsed within 2 years). Clinical factors, including age, number of platinum drug courses, platinum-free interval, and CA125 value before olaparib treatment, were compared between the responder and non-responder groups. **Results**: The age of the responder group was significantly younger than that of the non-responder group (52 vs. 69 years old, *p* = 0.02). The CA125 value of the responder group was significantly lower than that of the non-responder group (14.2 vs. 82.7 U/mL, *p* = 0.02). **Conclusions**: The good predictive factors that enabled continued olaparib administration without recurrence were younger age and a lower CA125 value before olaparib treatment. The younger group (<65 years old) and the low CA125 value group (<20 U/mL) in PSR may be treated with olaparib for a long period, suppressing disease progression. Providing this information to patients with PSR may help in decision-making regarding performing maintenance therapy with olaparib.

## 1. Introduction

In Japan, 12,738 women were diagnosed with ovarian cancer in 2019, and 5182 women died from the disease in 2022 [1]. The Ovarian Cancer Treatment Guidelines [2] recommend combination therapy with platinum and paclitaxel as the primary treatment for advanced and recurrent ovarian cancer. Approximately 70% of ovarian cancers are serous ovarian cancer [3]. Furthermore, 51% of serous ovarian cancers are diagnosed as clinically advanced stage III, with a 5-year survival rate of 42%, and the histological type has a poor prognosis [4].

Surgery plus postoperative chemotherapy (platinum) is the standard treatment for serous ovarian cancer [2]. Recurrences within 6 months of platinum therapy completion are frequently platinum-resistant, whereas recurrences >6 months later are often platinum-sensitive [5]. Therefore, platinum-based chemotherapy is indicated for patients with recurrent disease after ≥6 months or more [6,7,8]. Olaparib is indicated after platinum therapy in patients with platinum-sensitive recurrent ovarian cancer (PSR) as a treatment for recurrence [2]. If platinum agents are ineffective for recurrent ovarian cancer, the patient is not a candidate for treatment with poly ADP ribose polymerase inhibitors, including olaparib, and is treated as a second-line monotherapy (gemzar, Doxil, irinotecan, etc.) [9,10,11].

A meta-analysis of seven randomized controlled trials in PSR has revealed that olaparib extends patients’ overall survival [12]. Platinum resistance tends to be acquired, and recurrence occurs more quickly in PSR after platinum treatment completion [13]. Olaparib has provided long-term disease control in some cases, even after platinum therapy [14]. To date, a few reports from overseas focused on the characteristics of cases in which olaparib has achieved long-term disease control [14,15]. We decided to examine the clinical characteristics of patients who can take olaparib long-term in PSR.

The CA125 positivity rate is higher in serous ovarian cancer than in other histological types, making CA125 useful for diagnosis and treatment monitoring [16]. Furthermore, in PSR, it has been suggested that the serum CA125 level at the start of olaparib administration may predict progression-free survival (PFS) as an effect of maintenance therapy using olaparib for the treatment of recurrence [17].

However, few reports have presented data on Japanese patients. Moreover, only a few reports described the association between CA125 and the efficacy of olaparib treatment.

Therefore, this study aimed to determine clinical characteristics, including CA125, in PSR cases treated with olaparib.

## 2. Materials and Methods

This study included 24 patients with recurrent ovarian cancer treated with olaparib for PSR at Toho University Medical Center Sakura from April 2018 to December 2023 (Figure 1). We retrospectively reviewed medical records and analyzed the effect of clinicopathological factors to predict the treatment efficacy of olaparib.

We investigated the patients’ backgrounds and the incidence of related adverse events in all 24 PSR cases. Prognosis analysis excluded 8 cases because they had not relapsed and continued treatment with olaparib for within 2 years, and thus this study included 16 cases (Figure 1). We categorized the 16 cases into 5 cases who continued taking medication for >2 years (responder group) and 11 cases who relapsed and discontinued within 2 years (non-responder group) (Figure 1). A *t*-test was conducted to compare the following data between the responder and non-responder groups: (1) age, (2) number of platinum drug regimens, (3) platinum-free interval, (4) CA125 value during recurrence, (5) CA125 value before starting olaparib, (6) rate of decline in CA125, and (7) blood biochemistry test (C-reactive protein, neutrophil count, lymphocyte count, lactate dehydrogenase (LDH) value, albumin, and calcium value).

Differences between groups were analyzed with *t*-tests for the categorical variables. A *p*-value of <0.05 indicated a significant difference. Overall survival (OS) and progression-free survival (PFS) curves were generated using the Kaplan–Meier method and assessed using univariate analyses, with differences evaluated using the log-rank test. *p*-values of <0.05 were considered statistically significant. These statistical analyses were conducted using JMP Pro 17. The Ethics Committee of Toho University Medical Center Sakura approved this study (approval number S23060).

## 3. Results

### 3.1. Background of 24 Patients Who Were Treated with Olaparib on PSR

Table 1 shows the patients’ backgrounds. The median age was 67 years (range: 46–84 years). Of the 24 cases, 21 had high-grade serous carcinoma, 2 had unclassified adenocarcinoma, and 1 had clear cell carcinoma. BRCA testing was conducted in 5 cases, with 2 positives (BRCA1: 1 case and BRCA2: 1 case) and 3 negatives. However, 19 cases were not tested. The clinical stages were IC (1 case), IIIA (5 cases), IIIB (2 cases), IIIC (13 cases), IVA (2 cases), and IVB (1 case). The median platinum-free interval was 9 months (range: 6–47 months). The median number of chemotherapy regimens before starting olaparib was 2 (range: 2–6). The median duration of olaparib treatment was 7 months (range: 1–55 months). Twenty-three of the 24 cases were in clinical stage III or higher and were at high risk of recurrence. The median PFS with olaparib was 7 months, and the prognosis for PSR was poor. However, some cases were able to take olaparib for a long period of time without recurrence.

### 3.2. Adverse Events During Olaparib Treatment

Table 2 presents adverse events in 24 cases treated with olaparib. Adverse events of grade ≥ 3 were observed in 10 (41.7%) patients. The main side effects were nausea and fatigue in 10 (41.7%) patients and anemia in 15 (62.5%) patients, of which 8 (33.3%) were grade ≥ 3. No patients discontinued olaparib due to adverse events.

### 3.3. Progression-Free Survival After Initiation of Olaparib Treatment (n = 16)

Figure 2 illustrates PFS for 16 cases at our hospital. The median PFS was 7.7 months. All recurrence cases occurred within 24 months, and no recurrence was observed in patients taking olaparib for >24 months.

#### 3.3.1. Comparison of the Background and Blood Chemistry Test Between Responder Group (n = 5) and Non-Responder Group (n = 11)

Clinicopathological characteristics are compared between the responder and non-responder groups (Table 3). No statistically significant differences were found in the number of chemotherapy regimens, platinum-free interval, CA125 values during recurrence before chemotherapy, or the rate of decline in CA125 related to chemotherapy. The median age was significantly younger in the responder group than in the non-responder group (52 years old versus 69 years old, *p* = 0.02). The median CA125 value was significantly lower in the responder group than in the non-responder group (14.2 U/mL versus 82.7 U/mL, *p* = 0.02). Blood test values were compared between the responder and non-responder groups (Table 4). C-reactive protein, neutrophils, lymphocytes, LDH, albumin, and calcium immediately before taking olaparib exhibited no differences between the two groups.

#### 3.3.2. Correlation Between CA125 and Age Just Before Taking Olaparib

The group who were able to take olaparib for >2 years were significantly younger and had significantly lower CA125 levels. Therefore, we examined whether there was a correlation between age and CA125 (Figure 3). A positive correlation was observed between age and CA125, with a correlation coefficient of 0.65.

### 3.4. Outcome

#### 3.4.1. Age at the Time of Starting Olaparib, Duration of Olaparib Treatment, and Survival Time

Figure 4 illustrates the duration of olaparib treatment and survival time after relapse and olaparib discontinuation, in order of age at the time of olaparib initiation. In patients aged ≥ 65 years, 7 of 8 patients died of recurrence, and 1 patient survived with cancer. Conversely, in cases aged < 65 years, 5 out of 8 cases have not experienced a recurrence and continued receiving olaparib, 1 case is alive and well with cancer, and 2 cases have died. The comparison of the OS between the two groups aged < 65 years and ≥65 years revealed significantly better OS in those aged < 65 years than in those aged ≥ 65 years (Log-rank test, *p* = 0.02) (Figure 5).

#### 3.4.2. Duration of Olaparib Treatment and Overall Survival Months

The duration of taking olaparib and the survival period after discontinuing olaparib are presented in the order of CA125 value (Figure 6). All four patients in the low CA125 value group (<20 U/mL) did not relapse and were continuing with olaparib. Of the 12 cases in the high CA125 value group (≥20 U/mL), 11 relapsed and discontinued olaparib, whereas 1 case continued olaparib.

We compared the OS between the low and high CA125 value groups (Figure 7). The low CA125 group demonstrated significantly better OS than the high CA125 group (*p* = 0.02).

## 4. Discussion

This study analyzed 16 PSR cases treated with olaparib and confirmed the clinical characteristics of effective cases. The younger (<65 years old) and low CA125 groups demonstrated significantly better OS than the other groups.

The median age of participants in Study 19 vs. L-MOCA trial was 58 years (range: 21–89 years) vs. 54 years (range: 50–61.5 years). All L-MOCA trial cases were aged < 65 years, and the participants were younger than in Study 19. Regarding the age, the median PFS of maintenance therapy with olaparib in Study 19 vs. L-MOCA trial was 8.4 vs. 16.1 months. L-MOCA trial revealed a longer PFS than Study 19 [18]. Additionally, a retrospective study at our hospital revealed that the median ages of the non-responder and responder groups were 69 years (46–83) and 52 years (46–64), respectively, with a *p*-value of 0.02, indicating a statistically significant difference. This study indicated that PFS with olaparib maintenance therapy for PSR is extended in younger patients. The PFS was significantly longer in the group aged < 65 years than in those aged ≥ 65 years at our hospital, so we considered that patients aged < 65 years could be treated with olaparib for a long period.

The cutoff value for CA125 in ovarian tumors is 35 U/mL [19]. On the one hand, regarding the cut-off value for CA125 in ovarian cancer, a study indicated that it should be set at 15–20 U/mL, the same as in postmenopausal women, since treatment is bilateral salpingo-oophorectomy [20]. Asana F et al. reported that the median PFS in the CA125 < 18 U/mL group before olaparib administration was 13 months, whereas the median PFS in the CA125 > 18 U/mL group was 6 months, showing a significant difference [17]. Our study revealed the CA125 value of 20 U/mL in four PSR cases immediately before taking olaparib, and all four cases had been recurrence-free for >2 years and continued taking olaparib. Additionally, OS was longer than in 11 cases with CA125 levels of ≥20 U/mL. These results indicate that CA125 of <20 U/mL in PSR immediately before taking olaparib after chemotherapy could be a guideline for patients who can take olaparib for >2 years.

In general, younger people tend to have higher response rates because they are more physically fit and can tolerate chemotherapy better, whereas older people may have lower response rates because they often have other health problems and are less able to tolerate the side effects of chemotherapy [21]. In our study, we found a positive correlation between age and CA125 just before taking olaparib in PSR. Because recurrent ovarian cancer requires more chemotherapy treatments, younger patients who can tolerate chemotherapy may have better tumor control.

A meta-analysis of 2406 patients (1497 in the treatment group and 909 in the control group) from seven randomized controlled trials of PSR reported on adverse events associated with olaparib [12]. Olaparib extended overall survival, but the incidence of adverse events (nausea, fatigue, vomiting, diarrhea, abdominal pain, headache, etc.) increased significantly [12]. Among the 24 PSR cases at our hospital, the incidence of adverse events of Grade 3 or higher was 42%, but there were no cases where treatment was discontinued due to adverse events, so we believe that it is fully possible to continue olaparib while controlling adverse events. Nausea occurred in 54% of patients and anemia in 76.4% of patients in the L-MOCA trial that targets Asians [10]. All 24 cases at our hospital were Asian, with nausea occurring in 41.7% of cases and anemia in 62.5% of cases, and the frequency of adverse events was closer to that of the L-MOCA trial than Study 19. Therefore, as an adverse event of olaparib, nausea may more likely occur in Caucasians, and anemia is more likely to occur in Asians. The frequency of adverse events in this study was comparable to known adverse events in Asian patients.

The number of cases in this study was limited, so it is necessary to increase the number of cases in the future and further investigate whether CA125 < 20 and age < 65 are appropriate conditions for continuing long-term olaparib in PSR.

## 5. Conclusions

The younger group (<65 years old) and the low CA125 value group (<20 U/mL) in PSR may be treated with olaparib for a long period and suppress disease progression. Providing this information to patients with PSR may help in decision-making regarding performing maintenance therapy with olaparib.

## Figures and Tables

**Figure 1 diseases-13-00051-f001:**
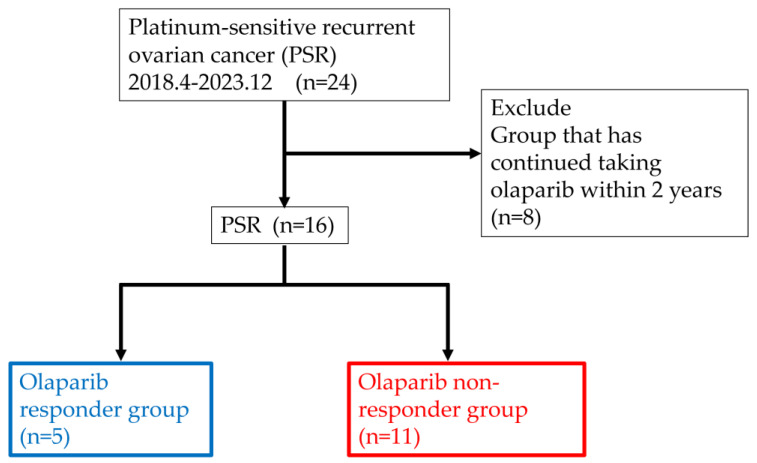
Patients’ selection for this study. PSR: platinum-sensitive recurrent ovarian cancer.

**Figure 2 diseases-13-00051-f002:**
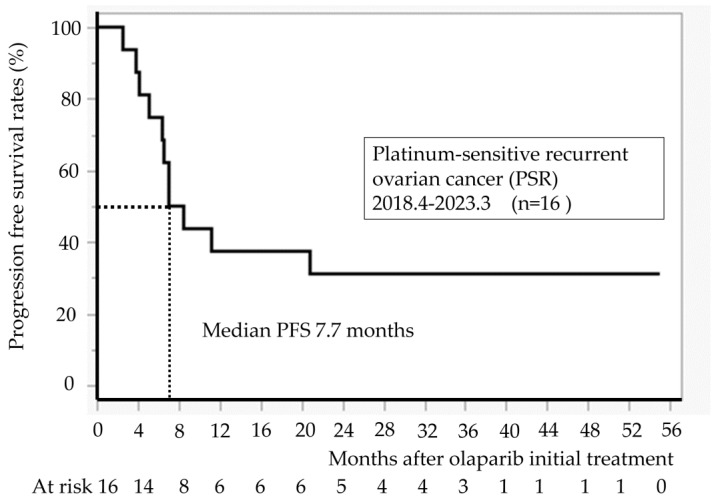
Progression-free survival after initiation of olaparib treatment (n = 16).

**Figure 3 diseases-13-00051-f003:**
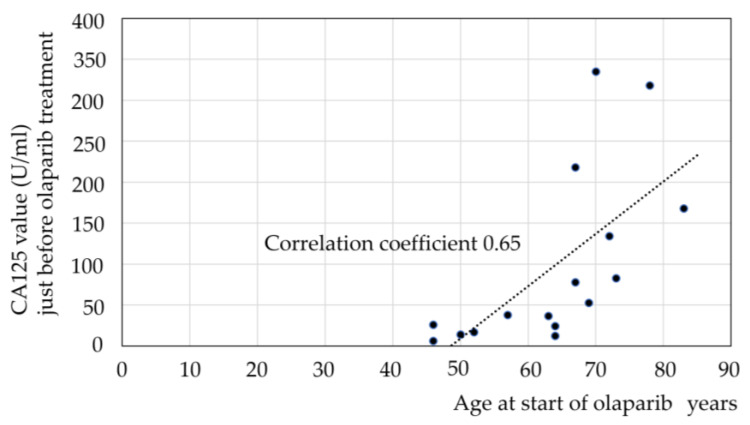
Correlation between CA125 and age. Correlation coefficient: 0.65. The dashed line shows the correlation coefficient.

**Figure 4 diseases-13-00051-f004:**
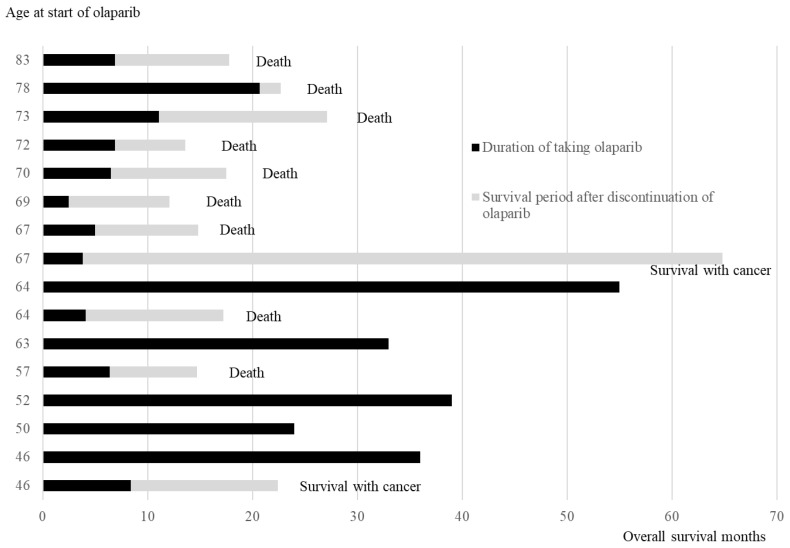
Age at the time of starting olaparib, duration of olaparib treatment, and survival time after relapse and discontinuation of olaparib.

**Figure 5 diseases-13-00051-f005:**
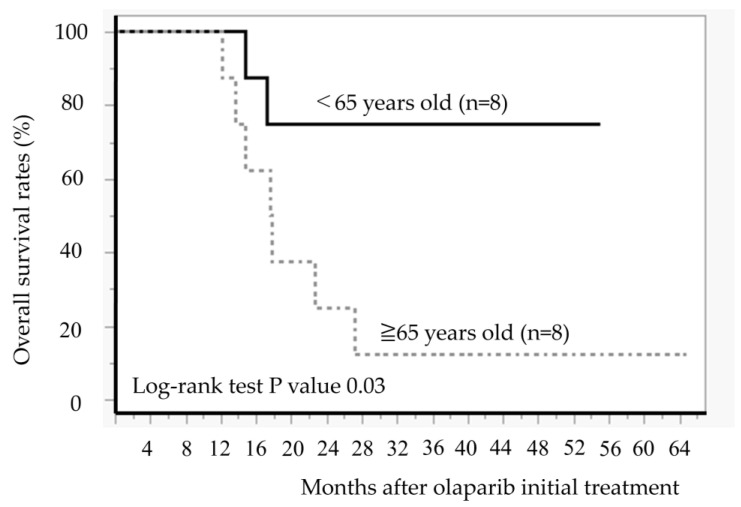
Comparison of overall survival after initiation of olaparib treatment between younger group (<65 years) and elderly group (≥65 years).

**Figure 6 diseases-13-00051-f006:**
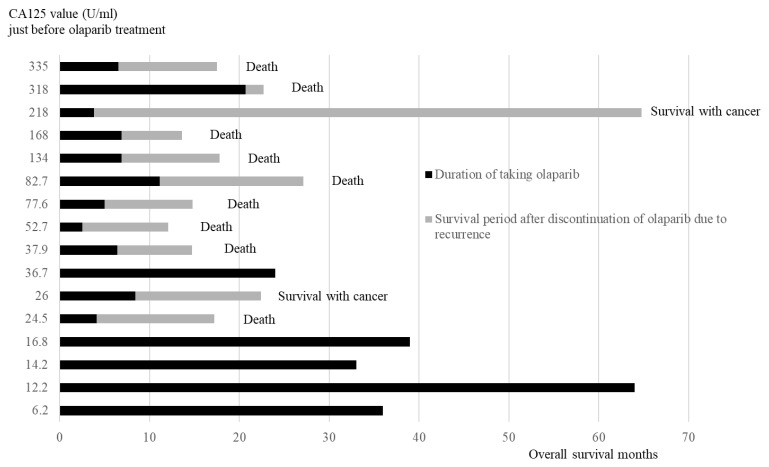
Duration of olaparib treatment and overall survival months after recurrence in order of CA125 values just before olaparib initial treatment.

**Figure 7 diseases-13-00051-f007:**
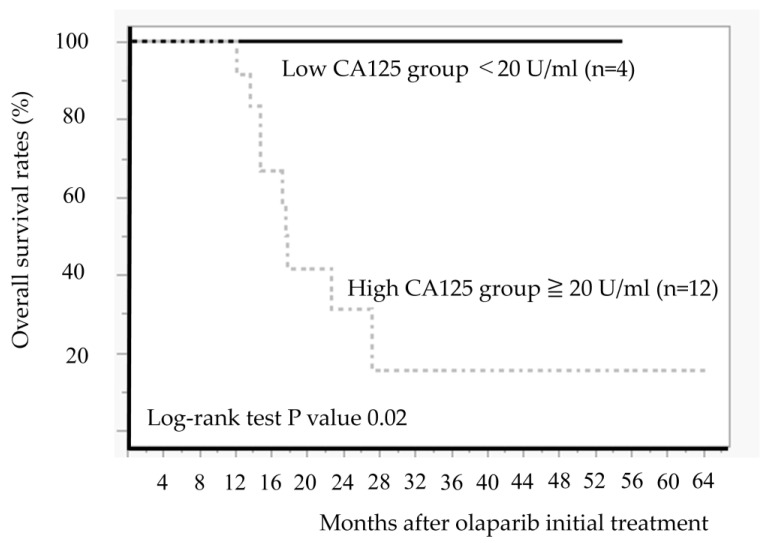
Comparison of overall survival after initiation of olaparib treatment between low CA125 group (<20 U/mL) and high CA125 group (≧20 U/mL).

**Table 1 diseases-13-00051-t001:** Background of 24 patients who were treated with olaparib.

Variables	
Age median (range)	67 (46–84) years
Histology type(serous/non serous)	21/3
gBRCA mutation (+/−/not inspected)	2/3/19
Clinical stage	IC:1 IIIA:5 IIIB:2 IIIC:13 IVA:2 IVB:1
Platinum free interval median (range)	9 (6–47) months
Number of chemotherapy regimens	2 (2–6)
Duration of taking olaparib median (range) months	7 (1–55)

gBRCA: germline breast cancer gene.

**Table 2 diseases-13-00051-t002:** Adverse events during olaparib treatment in 24 cases. Observation period: 1–55 months.

	All	Grade 1, 2	Grade ≤ 3
All adverse events of grade 3 or higher	10 (42%)	–	–
Nausea	10 (42%)	10 (42%)	0
General fatigue	10 (42%)	10 (42%)	0
Neutropenia	4 (17%)	2 (8%)	2 (8%)
Anemia	15 (63%)	7 (29%)	8 (33%)
Thrombocytopenia	3 (13%)	3 (13%)	0

**Table 3 diseases-13-00051-t003:** Comparison of the background. * *p* < 0.05.

	Olaparib for 2 Years or More (Respnder Group) Median (Range)	Olaparib Less Than 2 Years (Non-Responder Group) Median (Range)	*p* Value *
Duration of olaparib treatment (months)	36 (24–55)	7 (3–21)	
Age	52 (46–64)	69 (46–83)	0.02
Number of chemotherapy regimens	2 (2)	2 (2–6)	0.16
Platinum-free interval (months)	12 (9–36)	8 (6–30)	0.38
CA125 value (U/mL) at the time of recurrence	132 (9.5–132)	156 (62.8–8840)	0.14
CA125 value (U/mL) just before olaparib treatment	14.2 (6.2–36.7)	82.7 (24.5–335)	0.02
CA125 reduction rate due to chemotherapy	0.90 (0.35–0.92)	0.73 (−0.96–0.98)	0.24

**Table 4 diseases-13-00051-t004:** Comparison of blood tests and biochemical tests, pre-chemotherapy. * *p* < 0.05.

Blood Biomarkers Just Before Olaparib	Olaparib for >2 Years (Respnder Group) Median (Range)	Olaparib for <2 Years (Non-Responder Group) Median (Range)	*p* Value *
C-reactive-protein (mg/dL)	0.05 (0.01–1.64)	0.28 (0.01–1.98)	0.34
Neutrophils (μL)	2991 (1640–3392)	1808 (1230–6426)	0.31
Lymphocytes (μL)	1412 (600–1918)	1516 (659–2912)	0.37
LDH (U/mL)	209 (179–272)	192 (148–226)	0.08
Albumin (g/dL)	4.2 (3.4–4.5)	4.0 (3.6–4.6)	0.39
Calcium (mg/dL)	9.5 (9.2–9.9)	9.6 (9.2–9.9)	0.27

## Data Availability

Datasets generated and/or analyzed during the current study are available from the corresponding author upon reasonable request.

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
