# Peer review of "Clinical Impact of Olaparib for Platinum-Sensitive Recurrent Ovarian Cancer"

_diseases, 2025, doi:10.3390/diseases13020051_

Round 1
Reviewer 1 Report
Comments and Suggestions for Authors
This paper describes that the age and serum cancer antigen 125 (CA125) levels are good predictive factors for the efficacy of Olaparib maintenance therapy for patients with platinum-sensitive recurrent ovarian cancer. This is an interesting study. However, some sentences are not clear and concise. I recommend the following minor revision points to improve the quality of this article.
Minor points
1. Please check the paper title.
2. Please check the name of the affiliated institution. In the author information, it shows “Toho University Medical Center Sakura”, but in Materials and Methods section shows “Toho University Sakura Medical Center” and “Toho University Sakura Hospital Medical Center”.
3. Please remove the Arabic number from Keywords.
4. The experimental methodology states that the patient study period is from April 2018 to December 2023, however, Figure 1 shows the period from April 2018 to March 2023. Please check this point.
5. Please put a unit in X-axis in Figure 4.
6. The authors should check the format type, especially some technical words are typed in uppercase, and some are typed in lowercase letters.
7. Please remove the PSR (n=16) from Figure 6.
8. The authors should provide proper captions and footnotes in each Figures and Tables.
Comments on the Quality of English LanguageEnglish language and style are fine/minor spell check required.
Author Response
The corrections are marked in yellow.
Comment1. Please check the paper title.
Response1. The title has been changed.
Clinical impact of olaparib for platinum-sensitive recurrent ovarian cancer
→Retrospective study of cases in which olaparib can be continued long-term for platinum-sensitive recurrent ovarian cancer
Comment2. Please check the name of the affiliated institution. In the author information, it shows “Toho University Medical Center Sakura”, but in Materials and Methods section shows “Toho University Sakura Medical Center” and “Toho University Sakura Hospital Medical Center”.
Response2. Changed to Toho University Medical Center Sakura. Lines 73 and 93.
Comment3. Please remove the Arabic number from Keywords.
Response 3. The keywords have been modified as follows:
keyword 1; Ovarian cancer, 2; Olaparib, 3, Platinum-sensitive recurrent ovarian cancer, 4, CA125
→ Keywords: ovarian cancer, olaparib, platinum-sensitive recurrent ovarian cancer, CA125 Lines 34.
Comment 4. The experimental methodology states that the patient study period is from April 2018 to December 2023, however, Figure 1 shows the period from April 2018 to March 2023. Please check this point.
Response 4. Figure 1 changed to from April 2018 to December 2023.
(Figure1 2018.4-2023.12)
Comment 5. Please put a unit in X-axis in Figure 4.
Respons5.5.Months after initial olaparib treatment was inserted on the X-axis of Figure 4.
Comment 6. The authors should check the format type, especially some technical words are typed in uppercase, and some are typed in lowercase letters.
Response 6. Olaparib →olaparib
Comment 7. Please remove the PSR (n=16) from Figure 6.
Response 7. PSR (n=16) from Figure 6 was removed.
8. The authors should provide proper captions and footnotes in each Figures and Tables.
Footnotes were inserted on line 93 of Figure 1 and line 113 of Table 1.
PSR : platinum-sensitive recurrent ovarian cancer (line 93)
gBRCA : germline breast cancer gene (line 113)
Reviewer 2 Report
Comments and Suggestions for Authors Dear authors, thank you for effort on the manuscript titled Clinical Impact of Olaparib for Platinum-Sensitive 2 Recurrent Ovarian Cancer do address an important issue in cancer treatment, which will provide more insight for the ovarian cancer through the analysis toward the clinical ovarian patient, the analysis method in the research paper are appropriate. However, I would like to give a major revision based on below concerns : Major Issues o The introduction needs a clear focus to avoid confusion between platinum-sensitive and platinum-resistant cases. Restructure the introduction to consistently emphasize platinum-sensitive recurrent ovarian cancer. Avoid switching topics abruptly by clearly segmenting and transitioning discussions about different aspects of ovarian cancer treatment. For instance, if platinum resistance is mentioned, it should directly relate to explaining the context or rationale for focusing on platinum-sensitive cases in your study. o To provide a more informative introduction, consider incorporating key elements from the discussion that set the stage for your findings. For example, introduce the role and relevance of CA125 as a biomarker in the early part of your manuscript. This helps frame your research questions and objectives more comprehensively from the outset. o Ensure there is a logical flow in presenting results. For each table, provide a brief rationale for why this data is important and how it connects to your research questions. For example, explain how background data in Table 1 sets the stage for understanding the impact of Olaparib, and link the presentation of adverse events directly to the discussion of Olaparib's safety profile. o Address concerns about the adequacy of your sample size by discussing potential expansions or by clarifying the statistical power of your current sample. If increasing the sample size is not feasible, discuss the limitations this may impose on your findings and suggest how future studies could build on your work. Minor Issues o For Figure 1, enhance its visual appeal by adding color. Choose a palette that increases contrast between different data sections while ensuring the figure remains accessible to all viewers, including those who are colorblind. o For Figure 2 and other figures, improve readability by thickening the x and y axes. Ensure that the axis thickness is consistent across all figures to maintain a uniform appearance. o Check the legibility of all textual elements within the figures, such as labels, legends, and titles. Adjust font sizes as necessary and maintain a professional and consistent style across all figures. o Double-check each figure for any typographical errors or inaccuracies in data representation.Comments on the Quality of English Language
need improve
Author Response
Comment1. The introduction needs a clear focus to avoid confusion between platinum-sensitive and platinum-resistant cases. Restructure the introduction to consistently emphasize platinum-sensitive recurrent ovarian cancer. Avoid switching topics abruptly by clearly segmenting and transitioning discussions about different aspects of ovarian cancer treatment. For instance, if platinum resistance is mentioned, it should directly relate to explaining the context or rationale for focusing on platinum-sensitive cases in your study.
Rsponse1.
The following text has been added to the Introduction.
If platinum agents are ineffective for recurrent ovarian cancer, the patient is not a candidate for treatment with poly ADP ribose polymerase inhibitors, including olaparib, and is treated as a second-line monotherapy (gemzar, doxil, irinotecan, etc) [9-11] Lines 49-52
We decided to examine the clinical characteristics of patients who can take olaparib long-term in PSR. Lines 58-59
Comment 2. To provide a more informative introduction, consider incorporating key elements from the discussion that set the stage for your findings. For example, introduce the role and relevance of CA125 as a biomarker in the early part of your manuscript. This helps frame your research questions and objectives more comprehensively from the outset.
Response 2. The following text has been added to the Introduction.
Furthermore, in PSR, it has been suggested that the serum CA125 level at the start of olaparib administration may predict progression-free survival (PFS) as an effect of maintenance therapy using olaparib for treatment of recurrence [17]. Line2 62-64
Comment 3. Ensure there is a logical flow in presenting results. For each table, provide a brief rationale for why this data is important and how it connects to your research questions. For example, explain how background data in Table 1 sets the stage for understanding the impact of Olaparib, and link the presentation of adverse events directly to the discussion of Olaparib's safety profile.
Response 3. The following text has been added to the Materials and Methods and Discussion.
Twenty-three of the 24 cases were in clinical stage III or higher and were at high risk of recurrence. The median PFS with olaparib was 7 months, and the prognosis for PSR was poor. However, some cases were able to take olaparib for a long period of time without recurrence. Results Lime 107-110
A meta-analysis of 2,406 patients (1,497 in the treatment group and 909 in the control group) from seven randomized controlled trials of PSR reported on adverse events associated with olaparib [12]. Olaparib extended overall survival, but the incidence of adverse events (nausea, fatigue, vomiting, diarrhea, abdominal pain, headache, etc.) increased significantly [12]. Among the 24 PSR cases at our hospital, the incidence of adverse events of Grade 3 or higher was 42%, but there were no cases where treatment was discontinued due to adverse events, so we believe that it is fully possible to continue olaparib while controlling adverse events. Line2 209-216
Twenty-three of the 24 cases were in clinical stage III or higher and were at high risk of recurrence. The median PFS with olaparib was 7 months, and the prognosis for PSR was poor. However, some cases were able to take olaparib for a long period of time without recurrence.
Comment 4. Address concerns about the adequacy of your sample size by discussing potential expansions or by clarifying the statistical power of your current sample. If increasing the sample size is not feasible, discuss the limitations this may impose on your findings and suggest how future studies could build on your work.
Response 4. The following text has been added to the Discussion.
The number of cases in this study was limited, so it is necessary to increase the number of cases in the future and further investigate whether CA125 < 20 and age < 65 are appropriate conditions for continuing long-term olaparib in PSR. Discussion Lines 223-225
Comment 5. Minor Issues o For Figure 1, enhance its visual appeal by adding color. Choose a palette that increases contrast between different data sections while ensuring the figure remains accessible to all viewers, including those who are colorblind. o For Figure 2 and other figures, improve readability by thickening the x and y axes. Ensure that the axis thickness is consistent across all figures to maintain a uniform appearance. o Check the legibility of all textual elements within the figures, such as labels, legends, and titles. Adjust font sizes as necessary and maintain a professional and consistent style across all figures.
Response 5. Figure 1 has been colourised.
I made the X and Y axes thicker.(Figure 2,4,6)
All figures have been revised and adjusted.
Comment 6. Double-check each figure for any typographical errors or inaccuracies in data representation
Reponse 6. I checked the each figure .
Reviewer 3 Report
Comments and Suggestions for Authors
The MS entitled “Clinical Impact of Olaparib for Platinum-Sensitive Recurrent Ovarian Cancer” highlights two main features: 1) a higher survival of patients with low CA125, 2) a higher survival of younger patients. While the first evidence was previously described in a study of Asano et al., entitled “Serum CA125 level as predictors of the efficacy of olaparib maintenance therapy for platinum-sensitive relapsed ovarian cancer”, 2023, https://doi.org/10.1111/jog.15798. The second statement is obvious. While some new results are provided, the general significance of the study is obscure.
Several issues must be resolved:
1) A detailed comparison with a study of Asano et al.,
entitled “Serum CA125 level as predictors of the efficacy of olaparib maintenance therapy for platinum-sensitive relapsed ovarian cancer”, 2023, https://doi.org/10.1111/jog.15798
MUST BE PROVIDED!
2) The former study must be properly cited and discussed in the discussion section, and a statement that data on the impact of CA125 on prognosis were previously obtained by Asano et al.
3) 24 PSR cases were mentioned in the MS everywhere, while 8 cases were excluded from the analysis. Hence, 24 should be replaced with 16 cases to avoid confusion.
4) It is not clear if 8 of the 24 cases were excluded because they had been treated for <2 years (in the text) or continued treatment with olaparib for >2 years? (Fig. 1)
5) It was not clearly mentioned what was a reason for olaparib discontinuation. The respective statement must be provided.
6) There is clear evidence that duration of olaparib treatment influences survival. May olaparib discontinuation influence the results obtained?
Author Response
Comment 1. A detailed comparison with a study of Asano et al.,entitled “Serum CA125 level as predictors of the efficacy of olaparib maintenance therapy for platinum-sensitive relapsed ovarian cancer”, 2023, https://doi.org/10.1111/jog.15798 MUST BE PROVIDED!
Response 1. The following text has been added.
Furthermore, in PSR, it has been suggested that the serum CA125 level at the start of olaparib administration may predict progression-free survival (PFS) as an effect of maintenance therapy using olaparib for treatment of recurrence [17]. Introduction Line2 62-64
[17] Asano F, Momomura M, Morisada T, Tsushima K, Haruna Y, Shibuya H, Matsumoto H, Kobayashi Y. Serum CA125 level as predictors of the efficacy of olaparib maintenance therapy for platinum-sensitive relapsed ovarian cancer. J Obstet Gynaecol Res. 2023 Dec;49(12):2883-2888. doi: 10.1111/jog.15798. Epub 2023 Sep 21. PMID: 37735981.
Comment 2. The former study must be properly cited and discussed in the discussion section, and a statement that data on the impact of CA125 on prognosis were previously obtained by Asano et al.
Response 2. The following text has been added.
Asana F et al. reported that the median PFS in the CA125 < 18 U/mL group before olaparib administration was 13 months, whereas the median PFS in the CA125 > 18 U/mL group was 6 months, showing a significant difference [17]
Commet .3 24 PSR cases were mentioned in the MS everywhere, while 8 cases were excluded from the analysis. Hence, 24 should be replaced with 16 cases to avoid confusion.
Response 3. I changed the sentence ① to ②. Lines 75-77
①We investigated the duration of olaparib treatment and the incidence of related adverse events in all 24 PSR cases. Prognosis analysis excluded 8 of the 24 cases because they had been treated for <2 years, and thus this study included 16 cases (Figure 1).
→②We investigated the patients background and the incidence of related adverse events in all 24 PSR cases. Prognosis analysis excluded 8 cases because they had not relapsed and continued treatment with olaparib for within 2 years,
Comment .4 It is not clear if 8 of the 24 cases were excluded because they had been treated for <2 years (in the text) or continued treatment with olaparib for >2 years? (Fig. 1)
Response.4 Prognosis analysis excluded 8 cases because they had not relapsed and continued treatment with olaparib for within 2 years.
Commenr .5 It was not clearly mentioned what was a reason for olaparib discontinuation. The respective statement must be provided.
Response 5. I changed the sentence ① to ②. Lines 78-80
①We categorized the 16 cases into 5 cases who continued taking medication for >2 years (responder group) and 11 cases who relapsed within 2 years (non-responder group) (Figure 1).
②We categorized the 16 cases into 5 cases who continued taking medication for >2 years (responder group) and 11 cases who relapsed and discontinued within 2 years (non-responder group) (Figure 1).
Comment 6. There is clear evidence that duration of olaparib treatment influences survival. May olaparib discontinuation influence the results obtained?
Response 6. In PSR, if the CA125 level immediately before administration of olaparib is high or if the patient is elderly, there is a risk of relapse and discontinuation of olaparib even if administered.
Round 2
Reviewer 3 Report
Comments and Suggestions for Authors
1. The reply does not look like a detailed comparison suggested. It seems the MS may suffer from a lack of novelty.
2. The number of patients is rather low. It should be checked if there is a correlation between age and CA125 value just before olaparib treatment.
Author Response
Comment 1. The reply does not look like a detailed comparison suggested. It seems the MS may suffer from a lack of novelty.
Response 1. Thank you for your comment. We have examined the correlation between age and CA125 as suggested.
Comment 2. The number of patients is rather low. It should be checked if there is a correlation between age and CA125 value just before olaparib treatment.
Response 2. Thank you for your suggestions. Since we found a positive correlation between age just before taking olaparib and CA125, we added the following text and Figure 3.
3.4.2. Correlation between CA125 and age just before taking olaparib
The group who were able to take olaparib for >2 years were significantly younger and had significantly lower CA125 levels. Therefore, we examined whether there was a correlation between age and CA125 (Figure3). A positive correlation was observed between age and CA125 with a correlation coefficient 0.65. Lines 151-156
In general, younger people tend to have higher response rates because they are more physically fit and can tolerate chemotherapy better, whereas older people may have lower response rates because they often have other health problems and are less able to tolerate the side effects of chemotherapy [21]. In our study, we found a positive correlation between age and CA125 just before taking olaparib in PSR. Because recurrent ovarian cancer requires more chemotherapy treatments, younger patients who can tolerate chemotherapy may have better tumor control. Lines 216-222